# Links between Diet, Intestinal Anaerobes, Microbial Metabolites and Health

**DOI:** 10.3390/biomedicines11051338

**Published:** 2023-05-01

**Authors:** Sylvia H. Duncan, Elena Conti, Liviana Ricci, Alan W. Walker

**Affiliations:** Rowett Institute, University of Aberdeen, Foresterhill, Aberdeen AB25 2ZD, Scotland, UK; elena_conti@outlook.com (E.C.); alan.walker@abdn.ac.uk (A.W.W.)

**Keywords:** microbial symbiosis, host interactions, Firmicutes, Bacteroidetes, dietary macro-nutrients, short-chain fatty acids, butyrate, bacteriocins

## Abstract

A dense microbial community resides in the human colon, with considerable inter-individual variability in composition, although some species are relatively dominant and widespread in healthy individuals. In disease conditions, there is often a reduction in microbial diversity and perturbations in the composition of the microbiota. Dietary complex carbohydrates that reach the large intestine are important modulators of the composition of the microbiota and their primary metabolic outputs. Specialist gut bacteria may also transform plant phenolics to form a spectrum of products possessing antioxidant and anti-inflammatory activities. Consumption of diets high in animal protein and fat may lead to the formation of potentially deleterious microbial products, including nitroso compounds, hydrogen sulphide, and trimethylamine. Gut anaerobes also form a range of secondary metabolites, including polyketides that may possess antimicrobial activity and thus contribute to microbe–microbe interactions within the colon. The overall metabolic outputs of colonic microbes are derived from an intricate network of microbial metabolic pathways and interactions; however, much still needs to be learnt about the subtleties of these complex networks. In this review we consider the multi-faceted relationships between inter-individual microbiota variation, diet, and health.

## 1. Introduction

The human gastrointestinal tract harbours a diverse collection of bacteria, fungi, archaea, and viruses (microbiota). The human microbiota is most abundant in the large intestine, with the greatest microbial biomass accounted for by obligate anaerobes. The dynamics and activities of these anaerobes may be driven by a number of factors, including host genetics, microbial colonisation and succession after birth, and antibiotic usage. It is becoming increasingly apparent, however, that diet is a major driver in shaping the composition and activity of the gut microbiota [1], which in turn impacts on both gut and general health. Our indigenous gut microbial communities influence factors ranging from nutrient acquisition to protection against invading pathogenic microbes and the development and maintenance of the immune system [2,3]. The gut microbiota forms an assortment of primary and secondary metabolites, which may be functionally relevant in gut and systemic health [4,5,6,7,8,9,10].

Both microbiota composition and activity [11,12,13] can impact on host health at bodily sites both within and beyond the gastrointestinal tract, including via the gut–brain axis [14,15,16,17,18]. As such, there is now a greater appreciation than ever before of the proposed health benefits and risks associated with the makeup of our indigenous microbial companions, which may be mediated by diet [19]. Tremendous advances in the molecular methodologies to profile the intestinal microbiota have led to a good understanding of the most abundant species [20,21]. However, our knowledge of the impact of individual species on the stability and functionality of the microbial ecosystem is less well developed. Integrated approaches are therefore being used to better understand mechanisms behind health-altering microbial activities and to discover ways, including diet, to modulate the gut microbiota to promote health and to prevent disease [22].

The technology-driven increase in research activity has meant that knowledge is rapidly being gained about the importance of our indigenous intestinal microbial communities. Indeed, there is now accumulating evidence that links the gut microbiota composition and/or activity to a range of human diseases, many of which are common and cause significant health and economic burdens. For example, the resident microbiota may be an important factor in irritable bowel syndrome, which can affect approximately 20% of the population and which may be driven by changes in gut microbial composition [23]. The microbiota is also plausibly linked with disease activity in inflammatory bowel diseases (IBD) of ulcerative colitis (UC), where inflammation is confined to the colon, and of Crohn’s disease (CD), which can affect any part of the gastrointestinal tract [24]. Both diseases can be extremely debilitating, causing chronic diarrhoea, and abdominal cramping and pain, and incidences of both have risen as countries around the world have become more urbanised [25]. Colorectal cancer (CRC) is another ailment with potential microbiota involvement. Worldwide, almost two million cases are diagnosed annually [26], with diet and microbiota linked to up to 80% of colon cancers [27].

Given the importance of the topic and recent advances in the field, here, we focus on the compelling inter-relationships between diet, microbial composition/metabolism, and host health. We also highlight some of the emerging underlying mechanisms behind the interactions and links to health.

## 2. Composition and Activities of the Gut Microbiota

The colon is a largely anaerobic ecosystem, although there are oxygen gradients across the colonic wall, with free oxygen rapidly metabolised by facultative anaerobes [28]. The prevailing environmental conditions mean that most colonic bacteria are strict anaerobes. For example, some of these anaerobes, particularly those derived from the *Firmicutes* phylum such as *Roseburia* species, are extremely sensitive to oxygen and cannot survive more than two minutes of exposure to air [29].

### Human Gut Microbiota

The gut microbiota comprises species from all three domains of life (bacteria, archaea, and eukaryotes) as well as viruses, with each human individual harbouring a unique combination of constituent species. Bacteria are the most important contributors to microbiota biomass, with bacterial abundance progressively increasing along the intestinal tract. There are approximately 10^3^ cells per gram of contents in the duodenum of the small intestine, increasing up to approximately 10^10^ to 10^11^ bacterial cells per gram (wet weight) of faecal material [19]. Archaea are comparatively much less abundant in many individuals, although can reach densities of up to 10^9^ cells per gram of stool in some people [28]. Viruses are present at a density of at least 10^9^ particles per gram of faeces [30] with the bowel mucosa possessing a high virus to bacterial cell ratio [31], and most are bacteriophages [32]. As with gut bacteria, the viral populations exhibit inter-individual variability and are also diet-responsive, likely because viral population dynamics are heavily influenced by their prevailing bacterial hosts [33,34]. Mycobiome studies suggest that *Malassezia*, *Trichospora*, and *Candida* species are typically the most dominant eukaryotes in the gut, although cell numbers are much smaller than the bacterial component [35]. The abundance of *Candida* species has been shown to significantly correlate with the consumption of carbohydrate-rich foods [36,37].

The dominant bacterial populations are mainly derived from five phyla; the Gram-positive *Firmicutes* and *Actinobacteria*, and the Gram-negative *Bacteroidetes*, *Proteobacteria*, and *Verrucomicrobia* [28,38]. *Akkermansia muciniphila* is a dominant and widespread species belonging to the latter and is a key mucin-degrading bacterium [39]. The strictly anaerobic *Firmicutes* and *Bacteroidetes* lineages predominate in most healthy individuals. This relatively simple composition at the phylum level, however, is in stark contrast to an extraordinary level of diversity at the species and strain levels, with many thousands of species now known to be capable of colonising the human colon.

Although there is a very broad range across the human population, *Bacteroidetes* spp. may comprise about 30% of the total microbiota on average, with some individuals tending to be either *Bacteroides* or *Prevotella* dominant [40]. Commonly occurring species include *Bacteroides vulgatus*, *Bacteroides ovatus*, *Bacteroides uniformis*, and *Prevotella copri*. The human colonic *Firmicutes* species mainly belong to two phylogenetic groups: the *Lachnospiraceae* that includes abundant genera such as *Eubacterium Roseburia*, *Blautia,* and *Coprococcus*, and the *Ruminococcaceae* that encompasses *Faecalibacterium* and *Ruminococcus* species [41]. Other commonly reported genera that are typically found in lower abundance include *Actinobacteria* species such as *Bifidobacterium adolescentis* and *Collinsella aerofaciens*. Facultative anaerobes tend to be less dominant but typically include species such as *Escherichia coli* and sometimes the sulphate-reducing bacterial species *Desulfovibrio piger*. The most common and abundant archaeal species are methanogens, most often *Methanobrevibacter smithii* [28]. It is also worth noting that many of the most abundant species in the human colon are available in culture, which allows researchers to conduct detailed analysis of their physiology, although a larger proportion of the less dominant species remain either uncultured or are yet to be described [42,43,44,45].

When considering the collective genome content (or “metagenome”) of the microbiota, the inherent and daunting complexity becomes even more apparent. The human gut metagenome is estimated to contain many millions of unique genes, many of which are currently uncharacterised, and is at least two orders of magnitude more complex than the human genome [21]. It is apparent, however, that a relatively small number of dominant species are widespread amongst the human population and often numerically abundant [21]. For example, in many healthy subjects, two of the most abundant species are the butyrate producers, *Faecalibacterium prausnitzii* and *Eubacterium rectale* [46]. There is also likely to be strong interplay between the species that comprise the densely populated and complex microbial ecosystem of the colon. Some of these interactions are competitive, including competition for available macro- and micro-nutrients, but there are also many types of non-competitive interactions, such as cross-feeding and/or co-operative degradation of recalcitrant dietary substrates [13,47].

The composition of the microbiota is largely stable in adulthood, but fluctuations in response to lifestyle factors such as dietary consumption patterns are common. Within individuals, there are also a multitude of gastrointestinal micro-environments available for microbial colonisation, such as in the mucus layer covering the gut mucosa, on the surface of food particles in the gut lumen, or liquid-phase luminal contents [48,49,50]. For bacterial species to persist in the human large intestine, cells must be incredibly competitive and have the capacity to adapt to a wide range of gut environmental factors such as fluctuations in pH [51,52], bile salt concentrations [53], and availability of micro-nutrients, including iron [54], which is an essential co-factor for some bacteria [55]. Many bacteria also have specific vitamin requirements [56]. This includes the B vitamins such as biotin [B7], cobalamin [B12], folate [B9], pyridoxine [B6], riboflavin [B2], and thiamine [B1] that are needed to help drive enzyme-catalysed reactions [57]. Bacteria that generate growth factors such as vitamins are prototrophs, whereas bacteria that cannot produce these are auxotrophs and need to obtain these from dietary sources or by cross-feeding from other bacteria. For example, certain dominant gut bacteria directly depend on the presence of thiamine and riboflavin to generate butyrate. Soto Martin et al. (2020) investigated the requirement of 15 human butyrate-producing gut bacterial strains for a range of B vitamins and revealed that one of the most abundant butyrate-producing species in the colon, *Faecalibacterium prausnitzii*, was auxotrophic for most of the vitamins tested and is therefore reliant on cross-feeding with vitamin-producing strains [58]. Vitamin-related co-factors are metabolically expensive to produce and therefore it should not be surprising that these growth factors are shared between gut bacteria.

## 3. Metabolic Activity of the Gut Microbiota

As mentioned previously, the human gut metagenome possesses many more genes than the human genome, providing the host with a markedly increased functional capacity, particularly with respect to the breakdown of complex carbohydrates such as fibres [59]. A huge range of metabolites are therefore produced by the gut microbiota, and many of these can influence host physiology and health. Examples of health-related metabolites include short-chain fatty acids (SCFAs), lactate, secondary bile acids, lipids, vitamins, plant-derived flavonoids, amino acid and peptide metabolites, and trimethylamine [60]. Although the array of gut microbiota-derived metabolites is daunting, metabolomic and host-centric approaches are being used to investigate the impact of a range of bacterial metabolites on host health [6,50,61].

Despite the aforementioned variation in microbiota composition between individuals, there are a number of conserved functional activities. Of key importance among these is bacterial fermentation, whereby energy sources such as complex dietary polysaccharides are broken down anaerobically into SCFAs plus gases, including hydrogen, carbon dioxide, and, in some individuals, methane [62,63,64,65,66]. The major SCFAs detected in the large intestine are acetate, propionate, and butyrate [67,68]. Other organic acids such as lactate, succinate, formate, and valerate are also formed by gut bacteria but tend not to accumulate at high concentrations in healthy individuals, primarily due to bacterial cross-feeding and conversion to other SCFAs [52]. The total SCFA levels in faeces may reach levels of approximately 50–150 mM on average, with up to 95% of SCFAs absorbed by the colonic mucosa. Of the SCFAs, butyrate is absorbed preferentially by colonic epithelial cells [52], and the colonic epithelium derives as much as 60 to 70% of its energy needs from butyrate [69]. It has been estimated that individuals consuming a typical Western diet may produce a total of 0.5 to 0.6 M of SCFAs per day, which provides up to 5 to 10% of daily calorific requirements [70,71]. Bacterially produced SCFAs also exert substantial influence over host factors such as colonic pH [51,52], gut transit time, and the absorption of sodium, potassium, and water [72,73,74,75].

The type of fermentation acids generated differs between species within the gut microbiota. Therefore, the concentration and type of SCFAs in the colon of a given individual will be impacted by the underlying species content of their microbiota. This may have implications for host health as different fermentation acids have varied impacts on the host. Butyrate, for example, is largely thought of as beneficial due to its putative anti-inflammatory and anti-carcinogenic effects, whereas lactate has a range of potential deleterious impacts (see Section 5 below for further examples of the impacts of SCFAs on the host).

Many bacterial species in the colon form acetate as a fermentation end product; therefore, it is often the most abundant SCFA [28]. Cross-feeding between formate-producing species, such as *Ruminococcus bromii*, and acetogens (which are bacteria that generate acetate as an end product of fermentation), including *Blautia* species, may be another substantial factor in the high rates of acetate production in the gut [76]. Dominant propionate producers include *Bacteroides*, *Prevotella*, and some *Negativicutes* species [28,77,78]. Key butyrate-producing bacterial lineages belong to the *Firmicutes* phylum and include *Eubacterium rectale*, *Roseburia* spp., *Anaerobutyricum* (formerly *Eubacterium*) spp., *Anaerostipes* spp., and *Faecalibacterium prausnitzii* [79,80]. Separately, lactate is formed by many different bacterial species in the colon, including *Bifidobacterium* species. However, this acid does not tend to accumulate in the healthy adult colon as it can be utilised as a growth substrate by cross-feeders [77,81]. Of these cross-feeders, some *Firmicutes* such as *Anaerobutyricum* (formerly *Eubacterium*) and *Anaerostipes* species are common inhabitants of the healthy human colon, and can metabolise lactate to form butyrate [61,62]. In vitro studies employing stable isotopes have revealed that lactate cross-feeding by species such as *Anaerobutyricum* and *Anaerostipes* spp. makes an important contribution (approximately 20%) to the butyrate pool [82].

### 3.1. Plant Phenolics

Depending on dietary intakes, adults consume approximately 1 g per day of polyphenols, which are a diverse class of plant secondary metabolites with a range of postulated impacts on host health. Therefore, one of the other metabolic activities of gut bacteria that may be of importance for human health is that they are likely to have a major role in transforming these plant polyphenols (as reviewed in depth elsewhere [22]), as well as releasing these compounds during breakdown of fibrous plant material. The impact of these metabolites on host health are therefore intricately linked with the underlying composition of an individual’s gut microbiota. Certain phytochemicals may not be fully released from plant fibres if a given gut microbiota lacks primary degrading species that are able to breakdown the plant material in which the phytochemicals are bound up, and final metabolite pools will also be strongly influenced by the ability (or not) of the gut microbiota to transform these phytochemicals to other biologically active compounds, many of which are absorbed and enter systemic circulation [83]. Thus, it follows that the individual response to identical dietary interventions can vary greatly, depending on their baseline microbiota composition. This has potentially important ramifications for generic dietary advice guidelines and for the emerging field of personalised nutrition.

### 3.2. Gut Microbiota and Small Molecules

There are also further molecules produced by the gut microbiota with human health relevance, which include a multitude of small molecules such as ribosomally encoded and post translationally modified peptides (RiPPs), saccharides, amino acids, non-ribosomal peptides (NRPs), NRP-independent siderophores and polyketides (PKs), and PK–NRP hybrids. RiPPs include several subclasses such as bacteriocins, microcins, lantibiotics, and thiopeptides, which are antibacterial in nature and can therefore be important mediators modulating the interactions and competition between bacteria. By inhibiting the growth of certain types of bacteria, they may also help to protect the host from invading pathogens [84]. Examples of ribosomally synthesised antimicrobial products produced by bacteria include lantibiotics such as nisin O discovered from the human gut bacterium *Blautia obeum* A2-162 [85]. There is another category of small molecules known as polyketides (PKs) with potential to inhibit human pathogens. Metagenomic studies have revealed that gut bacteria may have the capacity to form polyketides (PK) and/or non-ribosomally synthesised peptides (NRP) and PK–NRP hybrids [86]. However, rather few of these small molecules have actually been isolated and characterised from human microbiota to date. Examples of previously described compounds include reutericyclins, which are a family of antimicrobial metabolites synthesised by PK–NRP enzymes such as in *Lactobacillus reuteri* that can kill the pathogen *Clostridioides difficile* at physiologically relevant concentrations [87]. In addition to putative antimicrobial activities, other secondary metabolites released by the gut microbiota, including polyketides, may possess anti-tumour and cholesterol-lowering properties [88]. The discovery of novel bacterial secondary metabolites in gut anaerobes is therefore an area of research worthy of further investigation and potential pharmaceutical development. However, this is an emerging area of research, with much still to be learned. It is still unclear, for example, to what extent changing host dietary consumption patterns might promote the growth of specific bacteria that can synthesise health-related small molecules or influence their production of these metabolically active metabolites.

## 4. Links between Diet, Microbiota, and Host Health

As indicated previously, members of the human gut microbiota interact with each other in various ways, including as competitors, predators, and symbionts. To determine how they interact with each other, and how this subsequently influences host health, is one of the most intriguing challenges in biology.

Of the factors that are considered to influence the composition and metabolic outputs of the human colonic microbiota, diet is one of the major driving forces, in particular, dietary macro-nutrient content as well as micro-nutrients, including vitamins. Dietary effects start from birth as the nutrition that babies receive impacts on the composition and activities of their intestinal microbiota. The intestinal tract of breast-fed babies is largely dominated by members of the *Bifidobacterium* genus, which appear to be exquisitely adapted to utilise non-digestible (at least to the host) sugars in breast milk, usually referred to as human milk oligosaccharides (HMOs) [89,90]. These HMOs therefore drive bifidobacterial population establishment and expansion in the colon [91,92]. Formula-fed babies, in contrast, usually possess a more complex gut microbiota that is more adult-like in composition [93]. The introduction of solid foods at weaning results in completely altered substrate availability in the colon and triggers the expansion of the obligately anaerobic bacterial groups such as the *Bacteroidetes* and *Firmicutes* that are typically dominant in adulthood and are able to breakdown and metabolise more complex polysaccharide sources [94].

Differences in colonic bacterial composition are mirrored by SCFA profiles, with generally higher overall SCFA concentrations observed in individuals from more rural societies, presumably reflecting increased rates of bacterial breakdown of dietary fibres in response to greater consumption levels by the host [95]. On a cautionary note, certain reported studies have made comparisons across large published data sets, most of which have employed different methods for processing samples for DNA extraction and have used different kits for DNA extraction and amplification procedures and analytical tools for data processing, and these may all influence comparative analysis [96,97]. Nonetheless, repeated observations strongly suggest that dietary influences are likely to be a major factor that drive microbiota composition and activity [98].

In addition, human volunteer studies that involve precise control over dietary intake have provided further evidence of the impact of host diet on the species composition of the gut microbiota over relatively short time intervals [99]. In contrast to the broad-scale changes observed when comparing individuals with disparate dietary consumption patterns (e.g., urbanised populations in high-income countries versus more rural/traditional living communities), the impact of short-term dietary interventions generally appears to be more limited and reversible [40]. Many dominant species show limited responses to such dietary switches, possibly indicative of an inherent capacity of many constituent gut microbiota species to scavenge a broader range of substrates. Nonetheless, short-term shifts can impact on the composition and metabolic output of the microbiota, with certain species being more diet-responsive than other, more generalist, members of the microbiota. David et al. (2014) [12], for example, showed that switching from a plant-based diet to one high in animal fats and protein led to a significant reduction in *Firmicutes* species and an increase in *Bacteroides* species. In broad agreement with this finding, diets low in total carbohydrate content result in low numbers of butyrate-producing *Firmicutes* species, which leads to greatly reduced faecal butyrate levels [100]. Furthermore, changing the major non-digestible carbohydrate substrates from wheat bran to resistant starch also led to rapid and reversible increases in proportional abundance of a small number of bacterial groups, including *Ruminococcus bromii*, in adult volunteers [19].

Variations in gut microbiota composition have also been reported in the elderly, particularly in those living in long-term care facilities [101,102]. These changes may result for several different reasons, including a reduction in intake of fibrous diets, dentition issues, or conditions such as diverticulitis that slow transit time through the gut. Such dietary changes are likely to result in a decrease in overall microbial diversity, often occurring in tandem with a reduction in potentially beneficial bifidobacteria and an increase in *Proteobacteria*, which have been linked to bowel disease (as reviewed in more detail elsewhere by Duncan and Flint) [103].

### 4.1. Macro-Nutrients Utilised by Gut Bacteria after Host Weaning

For microbes colonising the human colon, there are a range of different energy sources available. Typically, these will comprise dietary compounds that have escaped digestion by host enzymes further up the gastrointestinal tract, as well as endogenous host secretions such as mucin. Dietary compounds that are available for utilisation by the colonic microbiota include resistant starch, non-starch polysaccharides (NSP), oligosaccharides, and proteins [46]. The ability of intestinal microbiota to degrade these substrates will be impacted by factors such as particle size, cooking/preparation methods, and water solubility [104,105].

#### 4.1.1. Microbial Utilisation of Dietary Polysaccharides

As mammals do not possess the extensive enzyme repertoire needed to degrade most structural plant polysaccharides, they therefore rely on, to varying extents, their gut microbiota for their degradation [46]. On average, the main type of dietary polysaccharide available for bacteria in the large intestine is resistant starch (RS) [106,107] (see Table 1).

While starch is degraded by human enzymes in the small intestine, a significant fraction, particularly of high-amylose starches, can pass undigested into the colon (Englyst et al., 1987) [107]. Several bacterial species possess the ability to utilise RS, but starch type is known to influence bacterial degradation profiles. *Ruminococcus bromii* has been shown to be an important keystone species for RS degradation [19,105,108] as individuals who lack *R. bromii* in their microbiota are unable to fully degrade RS, which is then detectable in their faeces [19]. Recent work has shed light on the enzymology that underpins the success of *R. bromii* as an RS degrader; this species possesses a complex extracellular starch-degrading structure referred to as an “amylosome” [109]. Other human gut-dwelling bacterial species, including *Eubacterium rectale* along with *Bifidobacterium*, *Bacteroides*, and *Parabacteroides* species have also been shown to utilise various forms of RS [110,111,112].

Non-starch dietary polysaccharides (NSP) are the second major source of dietary polysaccharides that can be utilised by gut microbes and include cellulose, pectins, hemicelluloses (such as xylan), and inulin (Table 1). Many of these plant fibre structures are relatively complex, meaning a significant metabolic effort is likely required to degrade them. For example, Wisker et al. (1998) [113] observed total NSP fermentation of between 65.8 and 88.6% on mixed diets and, when looking at single substrates, showed fermentability of between 54.4% and 96.9%. Given the density of bacterial cells in the colon, there is likely to be considerable competition for most of these substrates, and culture and molecular-based studies have given some clues as to the predominant NSP-utilising bacteria in the human gut. For example, pivotal work by Salyers et al. [114,115], which mainly focused on *Bacteroides* species, revealed that many species could ferment NSPs. For example, *B. ovatus* and *B. eggerthii* were shown to ferment xylan while *B. ovatus* and *B. thetaiotaomicron* fermented pectin and polygalacturonate. The rapid increase in the number of bacterial genomes that have been sequenced, which includes many of the dominant species from the human colon, has allowed researchers to identify many genes encoding polysaccharide utilisation functions. The complete genome for *B. thetaiotaomicron*, for example, appears to encode over 170 enzymes involved in polysaccharide breakdown [59,116]. In *Bacteroides* species, these genes are usually organised as polysaccharide utilisation loci (PUL) [117] that include genes encoding esterases, glycosyl hydrolases, and outer membrane proteins responsible for the initial binding of soluble polysaccharides [118].

The Gram-positive *Firmicutes* have been less well studied, although they account for approximately 65–70% on average of the species within the human colonic microbiota [28]. This is most likely due to the fact that, thus far, they have proven to be more difficult to isolate, culture, and grow in the laboratory [13]. Nonetheless, molecular analyses have shown that *Firmicutes* lineages such *Roseburia* species, which belong to the *Lachnospiraceae* family, in addition to members of the *Ruminococcaceae* family, are important colonisers of insoluble plant fibres [50,119], and *Roseburia inulinivorans* is one of the few colonic bacterial species that has been demonstrated, thus far, to metabolise long-chain inulin [120].

Furthermore, *Monoglobus pectinolyticus*, *Lachnospira* (formerly *Eubacterium*) *eligens*, and *Faecalibacterium prausnitzii* are three of only a relatively small number of *Firmicutes* species that have been shown to extensively metabolise pectin [53,121]. Genome analyses suggest that *Firmicutes* tend to have fewer genes responsible for polysaccharide degradation than *Bacteroides* spp., which is likely to be as a consequence of their smaller genomes and their greater nutritional specialisation [121,122]. As a rule, *Firmicutes* may often be considered to be more specialist fibre degraders, with a limited preferred substrate range, whereas *Bacteroides* spp. may have more versatile and varied substrate utilisation repertoires. Inulin and oligosaccharides, such as fructo-oligosaccharides (FOS) and galacto-oligosaccharides (GOS), have commonly been used as prebiotics to promote bifidobacteria; however, these carbohydrate sources are also likely to support the growth of many other colonic anaerobes [91,123]. Therefore, as with phytochemical release discussed previously, the impact of consumption of complex dietary carbohydrates, on both the host and their resident indigenous microbiota, is likely to be at least partially dependent on the baseline microbiota composition within a given individual.

#### 4.1.2. Microbial Utilisation of Dietary Fats and Protein

Dietary fats are mainly metabolised to glycerol and are largely absorbed in the small intestine, with relatively small amounts, approximately 7%, reaching the large intestine [124]. In comparison with a large body of the literature examining the impact of dietary polysaccharides on the gut microbiota, much less is known about the influence of dietary fat. However, there is some evidence to suggest that these residual fats impact the gut microbiota composition, with high-fat diets associated with reduced microbial diversity, a reduction in the health-associated genus *Faecalibacterium*, and an increase in *Bacteroidetes* spp. [125]. High-fat diets are also likely to lead to an increase in bile acid secretion. Bile acids are synthesised from cholesterol in the liver and enter the duodenum where these acids can disrupt the lipid bilayer of bacterial membranes [126]. Some bacterial species may be more susceptible to the action of the bile acids than others, and this results in modulation of gut microbial composition with, for example, possible reductions in a number of bacterial species, including *Faecalibacterium prausnitzii* [53], a bacterium that has been reported to have potent anti-inflammatory activity [127]. Murine studies have also shown that diets high in saturated fats and cholesterol can alter the gut microbiota, which perturbs immune homeostasis [128,129]. An elevated inflammatory response has the potential to result in loss of intestinal integrity with a resultant leaky barrier, thereby allowing gut bacteria and bacterial products to translocate and potentially cause systemic disease [130].

Dietary proteins are also substrates for bacterial fermentation in the colon [131,132], and, depending on intake, approximately 3–18 g of dietary proteins enter the human large intestine every day (Table 1). Dietary proteins are hydrolysed into peptides and amino acids by both host- and bacteria-derived proteases and peptidases [133,134,135], with the resulting peptides and amino acids either directly assimilated into microbial protein or fermented to provide energy for bacterial growth [136]. The latter tends to occur most commonly in more distal regions of the human colon where carbohydrate sources are depleted, and the luminal pH is near neutrality. Amino acid fermentation largely results in the formation of ammonia and the major SCFAs acetate, propionate, and butyrate, along with the formation of branched-chain fatty acids (BCFAs), including *iso*-butyrate, 2-methylbutyrate, and *iso*-valerate [137]. These BCFAs are mainly formed from the branched-chain amino acids valine, iso-leucine, and leucine, respectively, and can be used as faecal markers for protein fermentation. Bacterial deamination of aromatic amino acids leads to the production of the phenolic compounds p-cresol, phenylpropionate (from tyrosine), phenylacetate (from phenylalanine), and indole propionate and indole acetate (from tryptophan) [138]. Predominant peptide and amino acid-fermenting bacteria in human faeces include species that belong to the *Bacteroidetes* and *Firmicutes* phyla. Of the known proteolytic *Bacteroides* species, the *B. fragilis* group appears to possess potent peptidase activity. Indeed, genome sequence analysis of *B. fragilis* strains reveals the presence of many putative peptidases [139,140].

It is important to note that, in contrast to fermentation of fibres/complex carbohydrates by the gut microbiota, which is associated with a range of health benefits, protein fermentation may be more deleterious for the host. High levels of dietary protein intake, particularly of red and processed meats, is associated with an increased risk of a range of diseases, including cancer and inflammatory conditions [141,142,143,144]. Moreover, consumption of high-fat and high-protein diets has been shown to result in increased levels of toxic bacterial metabolites [11]. These include nitrosamines, heterocyclic amines, faecapentanes, super oxide radicals, and hydrogen sulphide.

Amines have been associated with migraines, the onset of hypersensitive syndromes, and induction of portal-systemic encephalopathy or hepatic coma in liver disease [145]. Heterocyclic amines, in particular, can be further transformed to genotoxic intermediates both in the liver and in the colon [146], and these products are likely to be linked to increased risk of colorectal cancer (CRC). Colonic bacteria also have a role in forming N-nitroso compounds [147]. Levels of N-nitroso compounds have been shown to be elevated following intake of high-protein diets, particularly meat [11]. Dietary proteins, particularly meat, may also be a source of sulphur amino acids that may promote sulphate-reducing bacteria such as *Desulfovibrio piger*, which produce the metabolite hydrogen sulphide [148,149] that is toxic to colonocytes and inhibits butyrate oxidation, which is the major energy source for colonocytes [150].

Most colorectal malignancies occur in the distal colon, where bacterial SCFA levels are comparatively low. This indicates that, as more easily accessible fibre sources become depleted along the colon, the microbiota instead switches to putrefactive fermentation, leading to the hypothesis that the microbiota may have a part to play in the production of carcinogens, tumour promoters, and/or reduced formation of anti-carcinogens such as butyrate [151,152]. Furthermore, bacterial species correlated with CRC, such as invasive *Escherichia coli*, enterotoxigenic *Bacteroides fragilis* (ETBF), and *Fusobacterium nucleatum* are likely to use peptides and amino acids as sources of energy, although the potential mechanisms involved in CRC development are not fully understood. Alteration in barrier function, however, allows adherent-invasive *E. coli* to access the epithelium and release toxins into the host tissue [153,154]. More specifically, *E. coli* strains of phylogenetic group B2 produce the genotoxin colibactin, which is encoded from a polyketide synthase (pks) genotoxic island and enhances double-strand DNA breaks, cell cycle arrest, and chromosome aberrations [155]. Another toxin-producing gut bacterial taxon is the subgroup of *B. fragilis* that synthesises the zinc-containing metalloprotease toxin fragilysin, also called *B. fragilis* toxin (BFT), which is encoded by the *bft* gene [156,157,158,159]. BFT disrupts the association of E-cadherin with β-catenin, altering cellular morphology and physiology [160]. A third CRC-associated bacterium, *F. nucleatum*, produces a virulence factor, adhesin FadA, through which it adheres to and invades epithelial and endothelial cells [161].

Other microbial metabolites that are linked to high-protein diets include trimethylamine (TMA). This is generated by selected members of the colonic bacteria from metabolism of compounds found in fish, red meat, and eggs such as choline, betaine, trimethylamine N-oxide (TMAO), and carnitine. The microbially derived TMA is oxidised to TMAO by flavin monooxygenases in the liver, where it can then pass into the bloodstream and may contribute to cardiovascular disease (CVD) [162]. However, it remains unclear whether elevated TMAO levels are truly causative for CVD, and further research is required.

## 5. Diet–Microbiota–Immune Function Crosstalk

Many of the interlinks between diet, microbiota, and host health are mediated via the immune system. Microbe-associated molecular patterns (MAMPs) derived from gut microbiota cells, as well as metabolites arising from microbial production, conversion, and degradation of diet- and host-derived substrates, form part of a continuous signalling network with the host (Figure 1).

From birth, the intestinal microbiota is instrumental in the development of immune responses to exogenous triggers, as well as in shaping tolerance to innocuous antigens and preventing over-activation of the immune system. From the host’s perspective, a large array of innate immune receptors and adaptive immune responses have evolved to nurture and sustain this alliance. Molecules contained in the mother’s milk and colostrum (such as immune effectors, cytokines, and immunoglobulin A) act to prevent overgrowth of potentially pathogenic microbes by binding microbial antigens and to promote immune adaptation to the presence of commensal species. At the same time, oligosaccharides contained in breast milk, which are indigestible to the host, support the growth of the relatively benign *Bifidobacterium* species [163,164].

In adult life, the gut microbiota continues to influence host immunity, often in ways that are linked to the host’s dietary consumption patterns. For example, SCFAs, particularly butyrate, produced as a result of the fermentation of dietary fibres by the gut microbiota are known to contribute to the maintenance of immune effectors such as regulatory T cells (Tregs). Tregs modulate activation, proliferation, and cytokine production of T cells, increasing production of anti-inflammatory cytokines such as IL-10. This contributes to containing immune activation against common antigens derived from food and commensals [165]. Research from rodent models suggests that inoculation with specific butyrate-producing bacterial species can also result in immunomodulatory effects. For example, oral administration of selected strains belonging to the *Ruminococcaceae* and *Lachnospiraceae* families of the *Firmicutes* phylum, resulted in an increased resilience of the animals to colitis and allergic diarrhoea [166,167]. Furthermore, SCFAs contribute to preservation of the epithelial barrier homeostasis. In particular, butyrate was shown to promote epithelial barrier function by depleting oxygen levels near the epithelium and stabilising the hypoxia-inducible factor (HIF), a transcriptional factor known to be necessary for modulating barrier protection [168], inducing the expression of genes responsible for tight junctions, and stimulating production of mucin [169], which may fortify the protective mucus layer that overlies the colonic epithelium. Similarly, lactate, a fermentation product derived from lactic acid bacteria such as *Bifidobacterium* and *Lactobacillus* species, was shown to significantly accelerate intestinal epithelial cell proliferation and development [170].

In addition to their stimulation of IL-10 production, SCFAs can also help to restrain inflammation through several other distinct mechanisms. For example, butyrate can downregulate the expression of pro-inflammatory molecules (such as IL-6, IL-12, and NOS2) released by macrophages in vitro and in vivo via inhibition of histone deacetylases [171] and can suppress lipopolysaccharide (LPS)-induced NF-κB activation through G-protein-coupled receptor GPR109A signalling on the apical membrane of epithelial cells both in vitro and ex vivo [172]. Further circumstantial evidence for the potential importance of butyrate in restraining inflammatory responses in the gut comes from the repeated observation that patients with inflammatory bowel disease often harbour lower levels of butyrate-producing bacteria in their gut than control patients [173].

As well as having anti-inflammatory effects [174], SCFAs can also positively impact the host’s reaction to pathogenic bacteria. Indeed, butyrate and propionate improve colonisation resistance against invading pathogens by enhancing the long-term bacterial killing capacity of intestinal macrophages if they are present during cell differentiation [171]. This stimulatory effect was shown to be driven by inhibition of HDAC3 and production of antibacterial proteins such as calprotectin [175].

In addition to SCFAs, other microbiota-derived metabolites can also impact the host immune system. For example, members of the genus *Bifidobacterium* and *Lactobacillus* can synthesise vitamin B9, folate, which is essential for human health and was reported to play an important role in the expansion of Tregs in the small intestine [176]. Similarly, vitamin B3, niacin, which can also be produced by gut microbiota, can ligate GPR109A and thereby suppress intestinal inflammation [172]. Furthermore, GPR109A has also been reported to be silenced in colon cancer in humans and in a mouse model and to induce apoptosis in colon cancer cells when expression is re-established, highlighting the potential role of microbial metabolites in mediating the role as a tumour suppressor in the human colon [172,174].

## 6. Targeted Dietary Approaches to Modulate the Gut Microbiota and Improve Host Health

As the composition and activities of the gut microbiota may have major implications for human health, there has been a drive to alter the composition to a state where there are larger numbers of beneficial species and smaller numbers of detrimental species, or to alter bacterial metabolism so that more beneficial by-products are formed [177]. This is typically attempted by making changes to the host diet via intake of fibres, probiotics, prebiotics, and synbiotics.

As discussed in Section 3 and Section 4, increased overall consumption of fibres typically results in elevated colonic SCFA production, which may be beneficial to the host for the reasons outlined in Section 5. Individual dietary components, such as prebiotics, may also have some health benefits. Prebiotics are usually oligosaccharides or inulins that are provided as supplements in certain food products to modulate the composition of the gut microbiota to maintain and promote gut health. As defined by Gibson et al. (2017) [178], a prebiotic is ‘a substrate that is selectively utilised by host microorganisms conferring a health benefit’. For prebiotic compounds to be considered efficacious, they must remain undigested by the enzymes of the small intestine and must selectively stimulate growth of beneficial bacteria in the gut [178,179,180,181,182]. Traditionally, prebiotics have been targeted towards promotion of the *Bifidobacterium* species in particular. However, oligosaccharides and inulins can also be utilised by other bacterial groups, meaning there are typically “off-target” effects when using these substrates as prebiotics [120]. This means that responses to prebiotic interventions will be, to some extent, dependent on the baseline microbiota composition of the individual consuming them. It also means that further research is required to fully understand the health-related impacts of prebiotic interventions. Nonetheless, stimulation of beneficial bacteria by prebiotic substrates may provide a number of putative beneficial effects by protecting the gut mucosa from enteric infection by lowering the intestinal pH via SCFA production, suppressing putrefactive and pathogenic bacteria and stimulating the immune system [183]. Synbiotics may also be used, which are a mixture of probiotics and prebiotics. At the moment though, the evidence for the efficacy of prebiotics and synbiotics is inconsistent. This may be partly due to inter-individual variations in gut microbiota response, but also because there are large differences in the types of products consumed and how they are administered. There is no consensus on optimal doses or lengths of intervention or on optimal ages for interventions. Clearly, although an area that holds some promise, more research is required to establish optimal intervention strategies at population and individual levels.

## 7. Conclusions

Advances in molecular technologies have led to a rapid improvement in our understanding of the composition of the gut microbiota in health and disease. The ongoing developments in sequence-based methodologies, such as the ability to generate metagenome-assembled genomes (MAGs) from gut microbiota species that have yet to be cultured in the laboratory, mean that we can now study our microbial inhabitants and the interconnections between them in far greater detail than ever before [184]. We now better appreciate the extent to which changing diets may be altering our gut microbiota on a global scale [184] and how microbes are transmitted and evolve within hosts over time [185]. It is increasingly apparent, however, that these diverse microbial communities perform a myriad of complex microbe–microbe and host–microbe interactions, and much remains to be learned about the impact of these interactions on host health. The gut microbiota is clearly recognised as having important roles in regulating various human processes, including metabolism and immunity, with emerging evidence indicating roles in systemic host physiology, such as cardiovascular health, as well as brain and cognitive function. The advances in molecular methodologies to profile the gut microbiota, along with the range of analytical methods available to measure microbial metabolites, should allow scientists in the future to provide tangible outcomes to target health maintenance through dietary strategies in the very young, the elderly, the under-nourished, as well as for certain gut disorders. A deeper understanding of the composition and emerging activities of the gut microbiota will hopefully aid in the development of novel therapeutics and interventions.

## Figures and Tables

**Figure 1 biomedicines-11-01338-f001:**
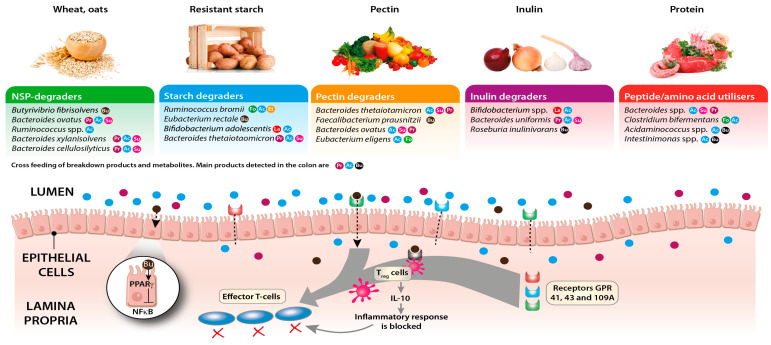
Links between diet, the intestinal microbiota, SCFAs, and inflammation. Predominantly anaerobic bacteria in the intestinal lumen ferment complex carbohydrates and proteins to shorter chain length breakdown products and short-chain fatty acids that bind to G-protein-coupled receptors on the colonic epithelium and on Treg cells and can block effector T cell responses. Butyrate, a major energy source for the colonocytes, may also elicit anti-inflammatory effects by blocking NF-κB activation. Different dietary components may promote the growth of distinct groups of bacteria (examples are given in the figure), with subsequent impacts on SCFA production rates and profiles. Coding for metabolites: Fo = formate, Ac = acetate, Pr = propionate, Bu = butyrate, Su = succinate, La = lactate, Et = ethanol.

**Table 1 biomedicines-11-01338-t001:** Energy sources available for bacterial fermentation in the human colon [46,62,104].

Substrate	Components	Structure	Amount Consumed (g Per Day)
Carbohydrates	**Resistant Starch (RS)**	**RS**—polymer of α1,4 linked glucose residues	10–40
	**Non-starch polysaccharides** (e.g., cellulose, xylan, pectin, inulin)	**Cellulose**—homopolymer of β1,4 linked glucose residues **Xylan**—polymer of β1,4 linked xylose residues with arabinose **Pectin**—complex polymer of α1,4 linked D-galacturonic acid or rhamnogalacturonan backbone with side-chain sugars of xylose, galactose, and arabinose	10–30
		**Inulin**—polymer of β 2,1 D-fructose residues with a terminal α1,2 D-glucose (DP > 10)	3–11
	**Oligosaccharides** (e.g., raffinose, stachyose, fructo-, xylo- and galacto-oligosaccharides)	**FOS**—polymer of β 2,1 D-fructose residues with a terminal α1,2 D-glucose (DP < 10)	2–10
	**Unabsorbed sugars** (e.g., lactose, sugar alcohols)	**Lactose**—dimer of β D-galactose (1,4) D-glucose	2–12
Proteins	**Unabsorbed dietary residues**	Mix of amino acids (meat protein should provide all required essential amino acids)	2–10
	**Endogenous secretions** (e.g., pancreatic enzymes and bile, mucus)	Mucins are high-molecular-weight glycoproteins consisting of a protein backbone, rich in serine and threonine, surrounded by O- and N-linked oligosaccharide side chains terminated by sialic acids or sulphate groups	2–8
Other	**Bacterial fermentation products** (cross-feeding)	Mainly formate, acetate, lactate, and succinate	

## Data Availability

Not applicable.

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
