# Peer review of "Links between Diet, Intestinal Anaerobes, Microbial Metabolites and Health"

_biomedicines, 2023, doi:10.3390/biomedicines11051338_

Round 1

Reviewer 1 Report

This is a fabulous review offering the current status of the field. A little more information on the use of WGS, NGS for the uncultivable bacteria would be a plus.

Author Response

Response to reviewer 1 comments:

We thank the reviewer for their kind comments. We have now added a short statement in the conclusions section regarding the use of sequencing technologies to assess the uncultivable bacteria. In addition, we have added two further references to support our additional comments in the text (reference numbers 184 and 185).

Reviewer 2 Report

Dear authors

This review paper examines the complex relationships between gut microbiota and diet. Its is very well written and a good contribution to the field. I recommend publication.

Just minor comments

Generally the review is very text based. More diagrams would strengthen the manuscript.

Figure 1 is too small and unreadable in parts. Suggest making it larger and go horizontal.

Conclusion section could be improved

Page 1 Line 10 remove "core" in inverted comments, unscientific

Author Response

Response to reviewer 2 comments:

Thanks to the reviewer for their comments.

We included one figure in the article as this is a complex figure that broadly encompasses the links between dietary macronutrients, the main bacterial species that have a role in metabolising key dietary components and their short chain fatty acids produced along with the impact of these weak acids on the host. We consider that this may be more valuable than presenting the information in several different figures. In addition, we included one table of information which we feel id valuable for the reader.

Figure 1 is now larger and should be more legible to the reader.

On page 1, the text has been modified and “core” now removed.

The conclusion section has now been revised and includes two additional references.

Reviewer 3 Report

The manuscript summarizes in a concise and complete form the current state of knowledge about the microbiome, diet and health relationships. The work is complete and, after being published in its current form, will resonate widely among researchers.

Author Response

Response to reviewer 3 comments:

Thanks for the kind comments.